# Unsupervised Deep Haar Scattering on Graphs

Xu Chen[1,2], Xiuyuan Cheng[2], and Stéphane Mallat[2]

[1]Department of Electrical Engineering, Princeton University, NJ, USA
[2]Département d'Informatique, École Normale Supérieure, Paris, France

## Abstract

The classification of high-dimensional data defined on graphs is particularly difficult when the graph geometry is unknown. We introduce a Haar scattering transform on graphs, which computes invariant signal descriptors. It is implemented with a deep cascade of additions, subtractions and absolute values, which iteratively compute orthogonal Haar wavelet transforms. Multiscale neighborhoods of unknown graphs are estimated by minimizing an average total variation, with a pair matching algorithm of polynomial complexity. Supervised classification with dimension reduction is tested on data bases of scrambled images, and for signals sampled on unknown irregular grids on a sphere.

## 1  Introduction

The geometric structure of a data domain can be described with a graph [11], where neighbor data points are represented by vertices related by an edge. For sensor networks, this connectivity depends upon the sensor physical locations, but in social networks it may correspond to strong interactions or similarities between two nodes. In many applications, the connectivity graph is unknown and must therefore be estimated from data. We introduce an unsupervised learning algorithm to classify signals defined on an unknown graph.

An important source of variability on graphs results from displacement of signal values. It may be due to movements of physical sources in a sensor network, or to propagation phenomena in social networks. Classification problems are often invariant to such displacements. Image pattern recognition or characterization of communities in social networks are examples of invariant problems. They require to compute locally or globally invariant descriptors, which are sufficiently rich to discriminate complex signal classes.

Section 2 introduces a Haar scattering transform which builds an invariant representation of graph data, by cascading additions, subtractions and absolute values in a deep network. It can be factorized as a product of Haar wavelet transforms on the graph. Haar wavelet transforms are flexible representations which characterize multiscale signal patterns on graphs [6, 10, 11]. Haar scattering transforms are extensions on graphs of wavelet scattering transforms, previously introduced for uniformly sampled signals [1].

For unstructured signals defined on an unknown graph, recovering the full graph geometry is an NP complete problem. We avoid this complexity by only learning connected multiresolution graph approximations. This is sufficient to compute Haar scattering representations. Multiscale neighborhoods are calculated by minimizing an average total signal variation over training examples. It involves a pair matching algorithm of polynomial complexity. We show that this unsupervised learning algorithms computes sparse scattering representations.

This work was supported by the ERC grant InvariantClass 320959.

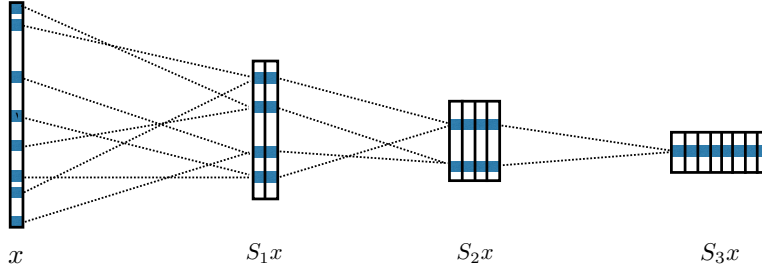

$$x \qquad S_1x \qquad S_2x \qquad S_3x$$

Figure 1: A Haar scattering network computes each coefficient of a layer $S_{j+1}x$ by adding or subtracting a pair of coefficients in the previous layer $S_jx$.

For classification, the dimension of unsupervised Haar scattering representations are reduced with supervised partial least square regressions [12]. It amounts to computing a last layer of reduced dimensionality, before applying a Gaussian kernel SVM classifier. The performance of a Haar scattering classification is tested on scrambled images, whose graph geometry is unknown. Results are provided for MNIST and CIFAR-10 image data bases. Classification experiments are also performed on scrambled signals whose samples are on an irregular grid of a sphere. All computations can be reproduced with a software available at *www.di.ens.fr/data/scattering/haar*.

## 2 Orthogonal Haar Scattering on a Graph

### 2.1 Deep Networks of Permutation Invariant Operators

We consider signals $x$ defined on an unweighted graph $G = (V, E)$, with $V = \{1, ..., d\}$. Edges relate neighbor vertices. We suppose that $d$ is a power of 2 to simplify explanations. A Haar scattering is calculated by iteratively applying the following permutation invariant operator

$$(\alpha, \beta) \longrightarrow (\alpha + \beta, |\alpha - \beta|) . \tag{1}$$

Its values are not modified by a permutation of $\alpha$ and $\beta$, and both values are recovered by

$$\max(\alpha, \beta) = \frac{1}{2}\big(\alpha + \beta + |\alpha - \beta|\big) \;\; \text{and} \;\; \min(\alpha, \beta) = \frac{1}{2}\big(\alpha + \beta - |\alpha - \beta|\big) . \tag{2}$$

An orthogonal Haar scattering transform computes progressively more invariant signal descriptors by applying this invariant operator at multiple scales. This is implemented along a deep network illustrated in Figure 1. The network layer $j$ is a two-dimensional array $S_jx(n, q)$ of $d = 2^{-j}d \times 2^j$ coefficients, where $n$ is a node index and $q$ is a feature type.

The input network layer is $S_0x(n, 0) = x(n)$. We compute $S_{j+1}x$ by regrouping the $2^{-j}d$ nodes of $S_jx$ in $2^{-j-1}d$ pairs $(a_n, b_n)$, and applying the permutation invariant operator (1) to each pair $(S_jx(a_n, q), S_jx(b_n, q))$:

$$S_{j+1}x(n, 2q) = S_jx(a_n, q) + S_jx(b_n, q) \tag{3}$$

and

$$S_{j+1}x(n, 2q + 1) = |S_jx(a_n, q) - S_jx(b_n, q)| . \tag{4}$$

This transform is iterated up to a maximum depth $J \leq \log_2(d)$. It computes $S_Jx$ with $Jd/2$ additions, subtractions and absolute values. Since $S_jx \geq 0$ for $j > 0$, one can put an absolute value on the sum in (3) without changing $S_{j+1}x$. It results that $S_{j+1}x$ is calculated from the previous layer $S_jx$ by applying a linear operator followed by a non-linearity as in most deep neural network architectures. In our case this non-linearity is an absolute value as opposed to rectifiers used in most deep networks [4].

For each $n$, the $2^j$ scattering coefficients $\{S_jx(n, q)\}_{0 \leq q < 2^j}$ are calculated from the values of $x$ in a vertex set $V_{j,n}$ of size $2^j$. One can verify by induction on (3) and (4) that $V_{0,n} = \{n\}$ for $0 \leq n < d$, and for any $j \geq 0$

$$V_{j+1,n} = V_{j,a_n} \cup V_{j,b_n} . \tag{5}$$

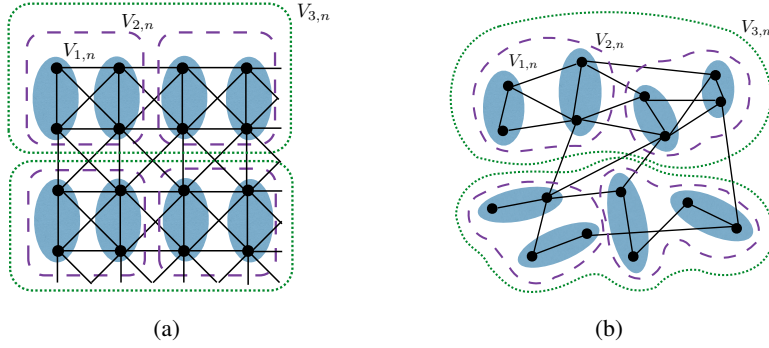

Figure 2: A connected multiresolution is a partition of vertices with embedded connected sets $V_{j,n}$ of size $2^j$. (a): Example of partition for the graph of a square image grid, for $1 \le j \le 3$. (b): Example on an irregular graph.

The embedded subsets $\{V_{j,n}\}_{j,n}$ form a multiresolution approximation of the vertex set $V$. At each scale $2^j$, different pairings $(a_n, b_n)$ define different multiresolution approximations. A small graph displacement propagates signal values from a node to its neighbors. To build nearly invariant representations over such displacements, a Haar scattering transform must regroup connected vertices. It is thus computed over multiresolution vertex sets $V_{j,n}$ which are connected in the graph $G$. It results from (5) that a necessary and sufficient condition is that each pair $(a_n, b_n)$ regroups two connected sets $V_{j,a_n}$ and $V_{j,b_n}$.

Figure 2 shows two examples of connected multiresolution approximations. Figure 2(a) illustrates the graph of an image grid, where pixels are connected to 8 neighbors. In this example, each $V_{j+1,n}$ regroups two subsets $V_{j,a_n}$ and $V_{j,b_n}$ which are connected horizontally if $j$ is even and connected vertically if $j$ is odd. Figure 2(b) illustrates a second example of connected multiresolution approximation on an irregular graph. There are many different connected multiresolution approximations resulting from different pairings at each scale $2^j$. Different multiresolution approximations correspond to different Haar scattering transforms. In the following, we compute several Haar scattering transforms of a signal $x$, by defining different multiresolution approximations.

The following theorem proves that a Haar scattering preserves the norm and that it is contractive up to a normalization factor $2^{j/2}$. The contraction is due to the absolute value which suppresses the sign and hence reduces the amplitude of differences. The proof is in Appendix A.

**Theorem 2.1.** *For any $j \ge 0$, and any $x, x'$ defined on $V$*

$$\|S_j x - S_j x'\| \le 2^{j/2} \|x - x'\| \,,$$

*and*

$$\|S_j x\| = 2^{j/2} \|x\| \,.$$

## 2.2 Iterated Haar Wavelet Transforms

We show that a Haar scattering transform can be written as a cascade of orthogonal Haar wavelet transforms and absolute value non-linearities. It is a particular example of scattering transforms introduced in [1]. It computes coefficients measuring signal variations at multiple scales and multiple orders. We prove that the signal can be recovered from Haar scattering coefficients computed over enough multiresolution approximations.

A scattering operator is contractive because of the absolute value. When coefficients have an arbitrary sign, suppressing the sign reduces by a factor 2 the volume of the signal space. We say that $S_J x(n, q)$ is a coefficient of order $m$ if its computation includes $m$ absolute values of differences. The amplitude of scattering coefficients typically decreases exponentially when the scattering order $m$ increases, because of the contraction produced by the absolute value. We verify from (3) and (4)

that $S_J x(n, q)$ is a coefficient of order $m = 0$ if $q = 0$ and of order $m > 0$ if

$$q = \sum_{k=1}^{m} 2^{J-j_k} \text{ for } 0 \le j_k < j_{k+1} \le J .$$

It results that there are $\binom{J}{m} 2^{-J} d$ coefficients $S_J x(n, q)$ of order $m$.

We now show that Haar scattering coefficients of order $m$ are obtained by cascading $m$ orthogonal Haar wavelet tranforms defined on the graph $G$. A Haar wavelet at a scale $2^J$ is defined over each $V_{j,n} = V_{j-1,a_n} \cup V_{j-1,b_n}$ by

$$\psi_{j,n} = 1_{V_{j-1,a_n}} - 1_{V_{j-1,b_n}} .$$

For any $J \ge 0$, one can verify [10, 6] that

$$\{1_{V_{J,n}}\}_{0 \le n < 2^{-J}d} \cup \{\psi_{j,n}\}_{0 \le n < 2^{-j}d, 0 \le j < J}$$

is a non-normalized orthogonal Haar basis of the space of signals defined on $V$. Let us denote $\langle x, x' \rangle = \sum_{v \in V} x(v) \, x'(v)$. Order $m = 0$ scattering coefficients sum the values of $x$ in each $V_{J,n}$

$$S_J x(n, 0) = \langle x , 1_{V_{J,n}} \rangle .$$

Order $m = 1$ scattering coefficients are sums of absolute values of orthogonal Haar wavelet coefficients. They measure the variation amplitude $x$ at each scale $2^j$, in each $V_{J,n}$:

$$S_J x(n, 2^{J-j_1}) = \sum_{\substack{p \\ V_{j_1,p} \subset V_{J,n}}} |\langle x , \psi_{j_1,p} \rangle|.$$

Appendix B proves that second order scattering coefficients $S_J x(n, 2^{J-j_1} + 2^{J-j_2})$ are computed by applying a second orthogonal Haar wavelet transform to first order scattering coefficients. A coefficient $S_J x(n, 2^{J-j_1} + 2^{J-j_2})$ is an averaged second order increment over $V_{J,n}$, calculated from the variations at the scale $2^{j_2}$, of the increments of $x$ at the scale $2^{j_1}$. More generally, Appendix B also proves that order $m$ coefficients measure multiscale variations of $x$ at the order $m$, and are obtained by applying a Haar wavelet transform on scattering coefficients of order $m - 1$.

A single Haar scattering transform loses information since it applies a cascade of permutation invariant operators. However, the following theorem proves that $x$ can be recovered from scattering transforms computed over $2^J$ different multiresolution approximations.

**Theorem 2.2.** *There exist $2^J$ multiresolution approximations such that almost all $x \in \mathbb{R}^d$ can be reconstructed from their scattering coefficients on these multiresolution approximations.*

This theorem is proved in Appendix C. The key idea is that Haar scattering transforms are computed with permutation invariants operators. Inverting these operators allows to recover values of signal pairs but not their locations. However, recombining these values on enough overlapping sets allows one to recover their locations and hence the original signal $x$. This is done with multiresolutions which are interlaced at each scale $2^j$, in the sense that if a multiresolution is pairing $(a_n, b_n)$ and $(a'_n, b'_n)$ then another multiresolution approximation is pairing $(a'_n, b_n)$. Connectivity conditions are needed on the graph $G$ to guarantee the existence of "interlaced" multiresolution approximations which are all connected.

## 3   Learning

### 3.1   Sparse Unsupervised Learning of Multiscale Connectivity

Haar scattering transforms compute multiscale signal variations of multiple orders, over non-overlapping sets of size $2^J$. To build signal descriptors which are nearly invariant to signal displacements on the graph, we want to compute scattering transforms over connected sets in the graph, which a priori requires to know the graph connectivity. However, in many applications, the graph connectivity is unknown. For piecewise regular signals, the graph connectivity implies some form of correlation between neighbor signal values, and may thus be estimated from a training set of unlabeled examples $\{x_i\}_i$ [7].

Instead of estimating the full graph geometry, which is an NP complete problem, we estimate multiresolution approximations which are connected. This is a hierarchical clustering problem [19]. A multiresolution approximation is connected if at each scale $2^j$, each pair $(a_n, b_n)$ regroups two vertex sets $(V_{j,a_n}, V_{j,b_n})$ which are connected. This connection is estimated by minimizing the total variation within each set $V_{j,n}$, which are clusters of size $2^j$ [19]. It is done with a fine to coarse aggregation strategy. Given $\{V_{j,n}\}_{0 \leq n < 2^{-j}d}$, we compute $V_{j+1,n}$ at the next scale, by finding an optimal pairing $\{a_n, b_n\}_n$ which minimizes the total variation of scattering vectors, averaged over the training set $\{x_i\}_i$:

$$\sum_{n=0}^{2^{-j-1}d} \sum_{q=0}^{2^j-1} \sum_i |S_j x_i(a_n, q) - S_j x_i(b_n, q)| \,. \tag{6}$$

This is a weighted matching problem which can be solved by the Blossom Algorithm of Edmonds [8] with $O(d^3)$ operations. We use the implementation in [9]. Iterating on this algorithm for $0 \leq j < J$ thus computes a multiresolution approximation at the scale $2^J$, with a hierarchical aggregation of graph vertices.

Observe that

$$\|S_{j+1}x\|_1 = \|S_j x\|_1 + \sum_q \sum_n |S_j x(a_n, q) - S_j x(b_n, q)| \,.$$

Given $S_j x$, it results that the minimization of (6) is equivalent to the minimization of $\sum_i \|S_{j+1}x_i\|_1$. This can be interpreted as finding a multiresolution approximation which yields an optimally sparse scattering transform. It operates with a greedy layerwise strategy across the network layers, similarly to sparse autoencoders for unsupervised deep learning [4].

As explained in the previous section, several Haar scattering transforms are needed to obtain a complete signal representation. The unsupervised learning computes $N$ multiresolution approximations by dividing the training set $\{x_i\}_i$ in $N$ non-overlapping subsets, and learning a different multiresolution approximation from each training subset.

## 3.2 Supervised Feature Selection and Classification

The unsupervised learning computes a vector of scattering coefficients which is typically much larger than the dimension $d$ of $x$. However, only a subset of these invariants are needed for any particular classification task. The classification is improved by a supervised dimension reduction which selects a subset of scattering coefficients. In this paper, the feature selection is implemented with a partial least square regression [12, 13, 14]. The final supervised classifier is a Gaussian kernel SVM.

Let us denote by $\Phi x = \{\phi_p x\}_p$ the set of all scattering coefficients at a scale $2^J$, computed from $N$ multiresolution approximations. We perform a feature selection adapted to each class $c$, with a partial least square regression of the one-versus-all indicator function

$$f_c(x) = \left\{ \begin{array}{ll} 1 & \text{if } x \text{ belongs to class } c \\ 0 & \text{otherwise} \end{array} \right. \,.$$

A partial least square greedily selects and orthogonalizes each feature, one at a time. At the $k^{th}$ iteration, it selects a $\phi_{p_k} x$, and a Gram-Schmidt orthogonalization yields a normalized $\tilde{\phi}_{p_k} x$, which is uncorrelated relatively to all previously selected features:

$$\forall r < k \,, \quad \sum_i \tilde{\phi}_{p_k}(x_i) \tilde{\phi}_{p_r}(x_i) = 0 \text{ and } \sum_i |\tilde{\phi}_{p_k}(x_i)|^2 = 1 \,.$$

The $k^{th}$ feature $\phi_{p_k} x$ is selected so that the linear regression of $f_c(x)$ on $\{\tilde{\phi}_{p_r} x\}_{1 \leq r \leq k}$ has a minimum mean-square error, computed on the training set. This is equivalent to finding $\phi_{p_k}$ so that $\sum_i f_c(x_i) \tilde{\phi}_{p_k}(x_i)$ is maximum.

The partial least square regression thus selects and computes $K$ decorrelated scattering features $\{\tilde{\phi}_{p_k} x\}_{k<K}$ for each class $c$. For a total of $C$ classes, the union of all these feature sets defines a dictionary of size $M = KC$. They are linear combinations of the original Haar scattering coefficients $\{\phi_p x\}_p$. This dimension reduction can thus be interpreted as a last fully connected network

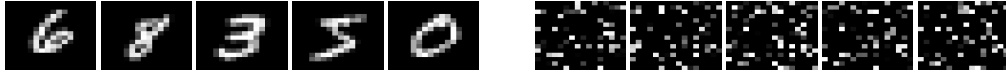

Figure 3: MNIST images (left) and images after random pixel permutations (right).

layer, which outputs a vector of size $M$. The parameter $M$ allows one to optimize the bias versus variance trade-off. It can be adjusted from the decay of the regression error of each $f_c$ [12]. In our numerical experiments, it is set to a fixed size for all data bases.

# 4 Numerical Experiments

Unsupervised Haar scattering representations are tested on classification problems, over scrambled images and scrambled data on a sphere, for which the geometry is therefore unknown. Classification results are compared with a Haar scattering algorithm computed over the known signal geometry, and with state of the art algorithms.

A Haar scattering representation involves few parameters which are reviewed. The scattering scale $2^J \leq d$ is the invariance scale. Scattering coefficients are computed up to the a maximum order $m$, which is set to 4 in all experiments. Indeed, higher order scattering coefficient have a negligible relative energy, which is below $1\%$. The unsupervised learning algorithm computes $N$ multiresolution approximations, corresponding to $N$ different scattering transforms. Increasing $N$ decreases the classification error but it increases computations. The error decay becomes negligible for $N \geq 40$. The supervised dimension reduction selects a final set of $M$ orthogonalized scattering coefficients. We set $M = 1000$ in all numerical experiments.

For signals defined on an unknown graph, the unsupervised learning computes an estimation of connected multiresolution sets by minimizing an average total variation. For each data basis of scrambled signals, the precision of this estimation is evaluated by computing the percentage of multiscale sets which are indeed connected in the original topology (an image grid or a grid on the sphere).

## 4.1 MNIST Digit Recognition

MNIST is a data basis with $6 \times 10^4$ hand-written digit images of size $d \leq 2^{10}$, with $5 \times 10^4$ images for training and $10^4$ for testing. Examples of MNIST images before and after pixel scrambling are shown in Figure 3. The best classification results are obtained with a maximum invariance scale $2^J = 2^{10}$. The classification error is $0.9\%$, with an unsupervised learning of $N = 40$ multiresolution approximations. Table 1 shows that it is below but close to state of the art results obtained with fully supervised deep convolution, which are optimized with supervised backpropagation algorithms.

The unsupervised learning computes multiresolution sets $V_{j,n}$ from scrambled images. At scales $1 \leq 2^j \leq 2^3$, $100\%$ of these multiresolution sets are connected in the original image grid, which proves that the geometry is well estimated at these scales. This is only evaluated on meaningful pixels which do not remain zero on all training images. For $j = 4$ and $j = 5$ the percentages of connected sets are respectively $85\%$ and $67\%$. The percentage of connected sets decreases because long range correlations are weaker.

One can reduce the Haar scattering classification error from $0.9\%$ to $0.59\%$ with a known image geometry. The Haar scattering transform is then computed over multiresolution approximations which are directly constructed from the image grid as in Figure 2(a). Rotations and translations define $N = 64$ different connected multiresolution approximations, which yield a reduced error of $0.59\%$. State of the art classification errors on MNIST, for non-augmented data basis (without elastic deformations), are respectively $0.46\%$ with a Gabor scattering [2] and $0.53\%$ with a supervised training of deep convolution networks [5]. This shows that without any learning, a Haar scattering using geometry is close to the state of the art.

| Maxout MLP + dropout [15] | Deep convex net. [16] | DBM + dropout [17] | Haar Scattering |
|:---:|:---:|:---:|:---:|
| 0.94 | 0.83 | **0.79** | 0.90 |

Table 1: Percentage of errors for the classification of scrambled MNIST images, obtained by different algorithms.

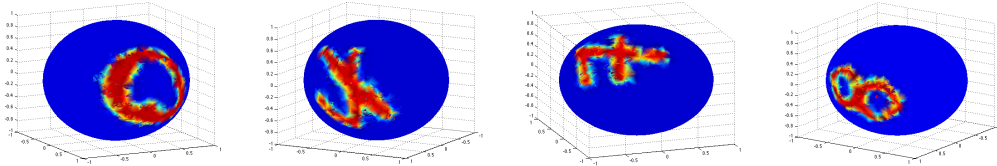

Figure 4: Images of digits mapped on a sphere.

## 4.2 CIFAR-10 Images

CIFAR-10 images are color images of $32 \times 32$ pixels, which are much more complex than MNIST digit images. It includes 10 classes, such as "dogs", "cars", "ships" with a total of $5 \times 10^4$ training examples and $10^4$ testing examples. The 3 color bands are represented with $Y, U, V$ channels and scattering coefficients are computed independently in each channel.

The Haar scattering is first applied to scrambled CIFAR images whose geometry is unknown. The minimum classification error is obtained at the scale $2^J = 2^7$ which is below the maximum scale $d = 2^{10}$. It maintains some localization information on the image features. With $N = 10$ multiresolution approximations, a Haar scattering transform has an error of $27.3\%$. It is $10\%$ below previous results obtained on this data basis, given in Table 2.

Nearly $100\%$ of the multiresolution sets $V_{j,n}$ computed from scrambled images are connected in the original image grid, for $1 \leq j \leq 4$, which shows that the multiscale geometry is well estimated at these fine scales. For $j = 5, 6$ and $7$, the proportions of connected sets are $98\%$, $93\%$ and $83\%$ respectively. As for MNIST images, the connectivity is not as precisely estimated at large scales.

| Fastfood [18] | Random Kitchen Sinks [18] | Haar Scattering |
|:---:|:---:|:---:|
| 36.9 | 37.6 | **27.3** |

Table 2: Percentage of errors for the classification of scrambled CIFAR-10 images, with different algorithms.

The Haar scattering classification error is reduced from $27.7\%$ to $21.3\%$ if the image geometry is known. Same as for MNIST, we compute $N = 64$ multiresolution approximations obtained by translating and rotating. After dimension reduction, the classification error is $21.3\%$. This error is above the state of the art obtained by a supervised convolutional network [15] ($11.68\%$), but the Haar scattering representation involves no learning.

## 4.3 Signals on a Sphere

A data basis of irregularly sampled signals on a sphere is constructed in [3], by projecting the MNIST image digits on $d = 4096$ points randomly sampled on the 3D sphere, and by randomly rotating these images on the sphere. The random rotation is either uniformly distributed on the sphere or restricted with a smaller variance (small rotations) [3]. The digit '9' is removed from the data set because it can not be distinguished from a '6' after rotation. Examples of the dataset are shown in Figure 4.

The classification algorithms introduced in [3] use the known distribution of points on the sphere, by computing a representation based on the graph Laplacian. Table 3 gives the results reported in [3], with a fully connected neural network, and a spectral graph Laplacian network.

As opposed to these algorithms, the Haar scattering algorithm uses no information on the positions of points on the sphere. Computations are performed from a scrambled set of signal values, without any

geometric information. Scattering transforms are calculated up to the maximum scale $2^J = d = 2^{12}$. A total of $N = 10$ multiresolution approximations are estimated by unsupervised learning, and the classification is performed from $M = 10^3$ selected coefficients. Despite the fact that the geometry is unknown, the Haar scattering reduces the error rate both for small and large 3D random rotations.

In order to evaluate the precision of our geometry estimation, we use the neighborhood information based on the 3D coordinates of the 4096 points on the sphere of radius 1. We say that two points are connected if their geodesic distance is smaller than 0.1. Each point on the sphere has on average 8 connected points. For small rotations, the percentage of learned multiresolution sets which are connected is 92%, 92%, 88% and 83% for $j$ going from 1 to 4. It is computed on meaningful points with nonneglegible energy. For large rotations, it is 97%, 96%, 95% and 95%. This shows that the multiscale geometry on the sphere is well estimated.

|  | Nearest Neighbors | Fully Connect. | Spectral Net.[3] | Haar Scattering |
|---|---|---|---|---|
| Small rotations | 19 | 5.6 | 6 | **2.2** |
| Large rotations | 80 | 52 | 50 | **47.7** |

Table 3: Percentage of errors for the classification of MNIST images rotated and sampled on a sphere [3], with a nearest neighbor classifier, a fully connected two layer neural network, a spectral network [3], and a Haar scattering.

## 5    Conclusion

A Haar scattering transform computes invariant data representations by iterating over a hierarchy of permutation invariant operators, calculated with additions, subtractions and absolute values. The geometry of unstructured signals is estimated with an unsupervised learning algorithm, which minimizes the average total signal variation over multiscale neighborhoods. This shows that unsupervised deep learning can be implemented with a polynomial complexity algorithm. The supervised classification includes a feature selection implemented with a partial least square regression. State of the art results have been shown on scrambled images as well as random signals sampled on a sphere. The two important parameters of this architecture are the network depth, which corresponds to the invariance scale, and the dimension reduction of the final layer, set to $10^3$ in all experiments. It can thus easily be applied to any data set.

This paper concentrates on scattering transforms of real valued signals. For a boolean vector $x$, a boolean scattering transform is computed by replacing the operator (1) by a boolean permutation invariant operator which transforms $(\alpha, \beta)$ into $(\alpha \text{ and } \beta, \alpha \text{ xor } \beta)$. Iteratively applying this operator defines a boolean scattering transform $S_j x$ having similar properties.

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
