[Supplementary Material]

# Unsupervised Deep Haar Scattering on Graphs Supplementary Material

## Appendix

## A    Proof of Theorem 2.1

*Proof.* Observe that the permutation invariant operator which associates to $(\alpha_0, \beta_0)$ the values

$$(\alpha_1, \beta_1) = (\alpha_0 + \beta_0, |\alpha_0 - \beta_0|)$$

satisfies

$$\alpha_1^2 + \beta_1^2 = 2(\alpha_0^2 + \beta_0^2).$$

Moreover, if $(\alpha_1', \beta_1') = (\alpha_0' + \beta_0', |\alpha_0' - \beta_0'|)$ then

$$(\alpha_1 - \alpha_1')^2 + (\beta_1 - \beta_1')^2 \leq 2\Big((\alpha_0 - \alpha_0')^2 + (\beta_0 - \beta_0')^2\Big).$$

Since $S_{j+1}x$ is computed by applying this operator to pairs of values of $S_j x$, we derive that

$$\|S_{j+1}x\|^2 = 2\|S_{j+1}x\|^2 \text{ and } \|S_{j+1}x - S_{j+1}x'\|^2 \leq 2\|S_j x - S_j x'\|^2.$$

Since $S_0 x = x$ and $S_0 x' = x'$, iterating on these two equations proves Theorem 2.1.

$\square$

## B    Haar Scattering from Haar Wavelets

The following proposition proves that order $m + 1$ scattering coefficients are computed by applying an orthogonal Haar wavelet transform to order $m$ scattering coefficients. We also prove by induction on $m$ that a scattering coefficient $S_j x(n, q)$ is of order $m$ if and only if $q = 2^j \kappa$ with

$$\kappa = \sum_{k=1}^{m} 2^{-j_k}$$

for some $0 < j_1 < ... < j_m \leq J$. This property is valid for $m = 0$ and the following proposition shows that if it is valid for $m$ then it is also valid for $m+1$ in the sense that an order $m+1$ coefficient is indexed by $\kappa + 2^{-j_{m+1}}$, and it is computed by applying an orthogonal Haar transform to order $m$ scattering coefficients indexed by $\kappa$.

**Proposition B.1.** *For any $v \in V$ and $0 \leq q < 2^j$ we write*

$$\overline{S}_j x(v, q) = \sum_{n=0}^{2^{-j}d-1} S_j x(n, q) \, 1_{V_{j,n}}(v).$$

*For any $\kappa = \sum_{k=1}^{m} 2^{-j_k}$, any $j_{m+1} > j_m$ and $0 \leq n < 2^{-j}d$,*

$$S_j x(n, 2^j(\kappa + 2^{-j_{m+1}})) = \sum_{\substack{p \\ V_{j_{m+1},p} \subset V_{j,n}}} |\langle \overline{S}_{j_m} x(\cdot, 2^{j_m}\kappa), \psi_{j_{m+1},p}\rangle|. \tag{B.1}$$

*Proof.* We derive from the definition of a scattering transform in equations (3,4) in the text that

$$S_{j+1}x(n, 2q) = S_j x(a_n, q) + S_j x(b_n, q) = \langle \overline{S}_j x(\cdot, q), 1_{V_{j+1,n}}\rangle,$$

$$S_{j+1}x(n, 2q+1) = |S_j x(a_n, q) - S_j x(b_n, q)| = |\langle \overline{S}_j x(\cdot, q), \psi_{j+1,n}\rangle|.$$

where $V_{j+1,n} = V_{j,a_n} \cup V_{j,b_n}$. Observe that

$$2^{j_{m+1}}(\kappa + 2^{-j_{m+1}}) = 2^{j_{m+1}}\kappa + 1 = 2(2^{j_{m+1}-1}\kappa) + 1,$$

thus $S_{j_{m+1}}x(n, 2^{j_{m+1}}(\kappa + 2^{-j_{m+1}}))$ is calculated from the coefficients $S_{j_{m+1}-1}x(n, 2^{j_{m+1}-1}\kappa)$ of the previous layer with

$$S_{j_{m+1}}x(n, 2^{j_{m+1}}(\kappa + 2^{-j_{m+1}})) = |\langle \overline{S}_{j_{m+1}-1}x(\cdot, 2^{j_{m+1}-1}\kappa), \psi_{j_{m+1},n}\rangle|. \tag{B.2}$$

Since $2^{j+1}\kappa = 2 \cdot 2^j \kappa$, the coefficient $S_{j_{m+1}-1}x(n, 2^{j_{m+1}-1}\kappa)$ is calculated from $S_{j_m}x(n, 2^{j_m}\kappa)$ by $(j_{m+1} - 1 - j_m)$ times additions, and thus

$$S_{j_{m+1}-1}x(n, 2^{j_{m+1}-1}\kappa) = \langle \overline{S}_{j_m}x(\cdot, 2^{j_m}\kappa), 1_{V_{j_{m+1}-1,n}}\rangle. \tag{B.3}$$

Combining equations (**??**) and (**??**) gives

$$S_{j_{m+1}}x(n, 2^{j_{m+1}}(\kappa + 2^{-j_{m+1}})) = |\langle \overline{S}_{j_m}x(\cdot, 2^{j_m}\kappa), \psi_{j_{m+1},n}\rangle|. \tag{B.4}$$

We go from the depth $j_{m+1}$ to the depth $j \geq j_{m+1}$ by computing

$$S_j x(n, 2^j(\kappa + 2^{-j_{m+1}})) = \langle \overline{S}_{j_{m+1}}x(\cdot, 2^{j_{m+1}}(\kappa + 2^{-j_{m+1}})), 1_{V_{j,n}}\rangle.$$

Together with (**??**) it proves the equation (**??**) of the proposition. The summation over $p$, $V_{j_{m+1},p} \subset V_{j,n}$ comes from the inner product $\langle 1_{V_{j_{m+1},p}}, 1_{V_{j,n}}\rangle$. This also proves that $\kappa + 2^{-j_{m+1}}$ is the index of a coefficient of order $m+1$. □

Since $S_0 x(n,0) = x(n)$, the proposition inductively proves that the coefficients at $j$-th level $S_j x(n, 2^j\kappa)$ for $j_m \leq j \leq J$ are of order $m$. The expression in the proposition shows that an $m+1$ order scattering coefficient at scale $2^J$ is obtained by computing the Haar wavelet coefficients of several order $m$ coefficients at the scale $2^{j_{m+1}}$, taking an absolute value, and then averaging their amplitudes over $V_{J,n}$. It thus measures the averaged variations at the scale $2^{j_{m+1}}$ of the $m$-th order scattering coefficients.

## C    Proof of Theorem 2.2

To prove Theorem 2.2, we first define an "interlaced pairings". We say that two pairings of $V = \{1, ..., d\}$

$$\pi^\epsilon = \{a_n^\epsilon, b_n^\epsilon\}_{0 \leq n < d/2}$$

are interlaced for $\epsilon = 0, 1$ if there exists no strict subset $\Omega$ of $V$ such that $\pi^0$ and $\pi^1$ are pairing elements within $\Omega$. The following lemma shows that a single-layer scattering operator is invertible with two interlaced pairings.

**Lemma C.1.** *Suppose that $x \in \mathbb{R}^d$ takes more than 2 different values, and two pairings $\pi^0$ and $\pi^1$ of $V = \{1, ..., d\}$ are interlaced, then $x$ can be recovered from*

$$S_1 x(n, 0) = x(a_n) + x(b_n), \quad S_1 x(n, 1) = |x(a_n) - x(b_n)|, \quad 0 \leq n < d/2.$$

*Proof.* By Eq. (2), for a triplet $n_1, n_2, n_3$ if $(n_1, n_2)$ is a pair in $\pi^0$ and $(n_1, n_3)$ a pair in $\pi^1$ then the pair of values $\{x(n_1), x(n_2)\}$ are determined (with a possible switch of the two) from

$$x(n_1) + x(n_2), \quad |x(n_1) - x(n_2)|$$

and those of $\{x(n_1), x(n_3)\}$ are determined similarly. Then unless $x(n_1) \neq x(n_2)$ and $x(n_2) = x(n_3)$ the three values $x(n_1), x(n_2), x(n_3)$ are recovered. The interlacing condition implies that $\pi^1$ pairs $n_2$ to an index $n_4$ which can not be $n_3$ or $n_1$. Thus, the four values of $x(n_1), x(n_2), x(n_3), x(x_4)$ are specified unless $x(n_4) = x(n_1) \neq x(n_2) = x(n_3)$. This interlacing argument can be used to extend to $\{1, \ldots, d\}$ the set of all indices $n_i$ for which $x(n_i)$ is specified, unless $x$ takes only two values. □

*Proof of Theorem 2.2.* Suppose that the $2^J$ multiresolution approximations are associated to the $J$ hierarchical pairings $(\pi_1^{\epsilon_1}, ..., \pi_J^{\epsilon_J})$ where $\epsilon_j \in \{0, 1\}$, where for each $j$, $\pi_j^0$ and $\pi_j^1$ are two interlaced pairings of $d2^{-j}$ elements. The sequence $(\epsilon_1, ..., \epsilon_J)$ is a binary vector taking $2^J$ different values.

The constraint on the signal $x$ is that each of the intermediate scattering coefficients takes more than 2 distinct values, which holds for $x \in \mathbb{R}^d$ except for a union of hyperplanes which has zero measure. Thus for almost every $x \in \mathbb{R}^d$, the theorem follows from applying Lemma **??** recursively to the $j$-th level scattering coefficients for $J - 1 \geq j \geq 0$. □