[Reviews · NeurIPS 2014]

Submitted by Assigned_Reviewer_17

The paper proposes an unsupervised learning adaptation of the scattering transform of Bruna and Mallat. Results are shown on permutation and rotated MNIST.

Clarity: I've implemented Bruna and Mallat's scattering transform, and I'm still not 100% sure whether Section 2 is the same or slightly different than it (because of the rephrasing as multirate filters, which I'm also familiar with). Both section 2 and 4 are very heavy-going: without sitting down and reconstructing the code, I don't think I really understand what is going on (esp. with Section 4). I understand that the authors are performing combinatorial optimization on a permutation of connections between layers in a scattering network. But, I don't understand the motivation for this, nor do I understand the details. I would encourage the authors to release their code, to assist in understanding.

Significance: The paper only tests the idea on MNIST and two artificially-altered versions of MNIST: permuted and 3D rotated. I find this to be not compelling enough for acceptance. Artificial permutation of data is not a naturally occuring transformation. Nor are digits naturally rotated in 3D. I would find the paper much more compelling if it were tested on natural data sets, such as CIFAR-10, or even more compelling: ImageNet.

I tried the original scattering transform on MNIST and CIFAR-10, and found that while it worked very well on MNIST, it did poorly on CIFAR-10. Thus, I am suspicious of any new scattering results that are not tested on natural image data sets.

I don't know if we're doing the authors a favor by accepting the paper in its current form (with weak results and difficult-to-understand motivation). If it turns out that this is a great idea, the authors will not be able to publish a more polished version with better results, because the reviewers of that hypothetical better paper will say it is incremental to this one.

Summary: The authors only show improvement on artifically altered versions of MNIST. I don't find this compelling enough for acceptance.

Submitted by Assigned_Reviewer_42

This paper seems very good but I do not understand the math.
Summary: Its clearly a good paper. I don't understand the mathematical details and I suspect that I never will.

Submitted by Assigned_Reviewer_43

This paper introduces a novel type of deep scattering network. It uses a cascade of wavelet transform convolutions and modulus non-linearities, adapted by permutations. What differentiates this work from previous scattering networks is that unsupervised learning is used to adapt the directions of contraction by maximizing the average discriminability (in terms of Euclidean distance) of training data. The method is shown to learn translation and rotation invariance on variants of MNIST.

Quality

The idea is novel and I found it interesting, even though I found it challenging to read as it is written from the perspective of signal processing rather than machine learning. I appreciate that the authors have presented the scattering transform as a product of multirate filter bank operators for the benefit of readers more familiar with convolutional neural networks. The paper sets out to highlight important principles governing the structure of convolutional neural networks (e.g. contraction) and relating them to standard signal processing tools; it achieves its goals. On the negative side, the experiments are not substantive and the paper ends abruptly without discussion or conclusion.

Clarity

The paper is readable despite the technically advanced material and non-mainstream topic. There are many small spelling and grammatical errors; it could certainly use more proofreading and polishing.

Originality

This is now one of a series of papers on deep scattering networks, but to the best of my knowledge, the first to pursue this idea of unsupervised learning by adapting permutations.

Significance

Deep scattering networks are attracting attention by the computer vision and machine learning communities (e.g. keynote by Stephane Mallat at CVPR 2014). This paper sheds more light on the connection between these models and convolutional neural networks, though I still wish this material was written to be more accessible to a wider audience.

Specific comments

The method is developed for the Haar family of wavelets; can you mention the implications of considering other families?

The paper develops a method for unsupervised contraction but doesn't reference any other works in unsupervised learning of representations, e.g. contractive autoencoders.
Summary: This paper presents an interesting variant of deep scattering networks that perform unsupervised learning by adapting permutations. The paper is missing some material (e.g. discussion, conclusion).

Submitted by Assigned_Reviewer_44

This paper uses transformations of the form |W|\pi to build deep networks, where W is a discrete Haar transform, | | is the pointwise absolute value nonlinearity, and \pi is a permutation of the index set of the input vector. The authors note that the \pi at each layer can be optimized for additive costs of pairs of indices in O(d^3) operations where d is the size of the input. They propose a cost function that maximizes the variance of the output of a layer. On the MNIST data set and a artificially transformed MNIST problem they use their (unsupervised) feature maps as inputs to an SVM and get strong classification results.

This is an interesting paper and should be accepted. The approach is unusual: they manage to arrange for a tractable combinatorial optimization for feature learning; after the optimization, the fprop is very fast. I can't think of another work that does this. On the other hand, the approach is not very well motivated. The authors use a abs nonlinearity, why? in their scattering network references, abs nonlinearities are used to introduce invariance to deformations; the choice of abs is well motivated because of its interactions with specially designed complex filters. Why do we expect an abs nonlinearity to reduce nuisance variation more than variation that we care about generically/unsupervised, and with real haar type filters? Moreover, the authors justify the cost function they use as the one that limits the contractiveness of their representation. What stops the reductions in contractiveness from being precisely reintroduction of variance in nuisance dimensions? The experiments do not give much intuition here: the MNIST data set is very special (it has well defined clusters), and previous works using sparse coding have gotten equally strong (permutation invariant) results (e.g. in the works of Julien Mairal and coauthors; although there the feature learning was supervised). I would have loved a page or two less about filter banks etc. and more experiments on other data sets, or more analysis of what is happening here.

minor details: citation 11 the authors names are mangled, on line 212 \pi^0 appears twice, 3.1 needs to have a more explicit/clearer definition of interlacing

All in all, interesting enough that it should really be accepted, but experiments are lacking: a strong result on MNIST is not especially illuminating or noteworthy.
Summary: This paper uses transformations of the form |W|\pi to build deep networks, where W is a discrete Haar transform, | | is the pointwise absolute value nonlinearity, and \pi is a permutation of the index set of the input vector. The authors note that the \pi at each layer can be optimized for additive costs of pairs of indices in O(d^3) operations where d is the size of the input. They propose a cost function that maximizes the variance of the output of a layer. On the MNIST data set and a artificially transformed MNIST problem they use their (unsupervised) feature maps as inputs to an SVM and get strong classification results.

The paper interesting enough that it should really be accepted, but experiments are lacking: a strong result on MNIST is not especially illuminating or noteworthy.
Author Feedback
Author rebuttal: We would like to thank the four reviewers for their careful evaluation of the paper, and their useful comments to improve it. We agree that the filter bank section makes the paper more difficult to read, and we shall thus concentrate on the Haar transform which is used in numerical applications. The filter bank extension will thus be reduced to a remark.

The main contribution of this paper is to provide a simple deep learning architecture, which can be optimised with exact polynomial algorithms, and be fully analysed mathematically (stability, contraction, energy conservation, inversion), while providing interesting numerical results. This is new, and we believe that it is needed to begin understanding the learning properties of these architectures.

Several reviewers rightfully mention that MNIST is not a highly challenging classification data basis. The MNIST data basis has been chosen in order to test performance of this algorithm, on a problem where the source of variations are well controlled: deformations, translations, rotations. It makes it possible to analyze the results in relation with the transform properties.

We agree with the reviewers that experiments on a more complex data basis such as CIFAR-10 would be a useful complement, that we shall thus try to include in the revised version. However, the path from mathematics to state of the art results on complex data bases takes time. A mathematical analysis of these highly non-linear deep architectures requires simplifications, while preserving key properties. Imposing close to state of the art results on complex data bases such as Imagenet would not allow to publish work devoted to mathematical understanding of such complex algorithms. Confrontations between theoretical and pratical algorithmic approaches are needed on this topic, and NIPS is a good place to do it.

Answers to reviewers questions:
- The absolute value non-linearity is justified for Haar wavelets by the fact that it provides a permutation invariant representations of pairs. It is in Section 3.1 but it will be explained before. The Haar scattering provides progressively more complex invariants by aggregating hierachically these elementary invariant pair representations.

- Haar wavelets are the only ones providing exact polynomial optimization algorithms, because they only group pairs of points. This will be better explained.

- References and comparisons with auto-encoders indeed need to be discussed.

- We will release the code.

- A final discussion of results will be incorporated as a conclusion